# Exposure of western United States bird communities to predicted high severity fire

Kari E. Norman [1,2] ✉, Andrew N. Stillman [3], Sean A. Parks [4,5], Courtney L. Davis [3] & Gavin M. Jones [1,2]

Fire is a pervasive biogeographic process that shapes biodiversity globally and is now experiencing unprecedented changes. Despite well documented impacts of fires on biodiversity, we do not know where biodiversity might be most vulnerable to changing fire regimes. We leverage recent advancements in fire forecasting and species distribution modeling to assess the exposure of bird species richness, community uniqueness, and functional richness to altered fire regimes in the western United States. We find that 55-58% of biodiversity hotspots are classified as "refugia", where high biodiversity intersects with predicted low severity burn areas. In contrast, 24-30% of biodiversity hotspots are classified as "areas of concern", where high biodiversity intersected with predicted high severity burn areas. Over half (52-60%) of "areas of concern" occur in geographies with historically low-severity fire regimes; a fire regime mismatch indicating a potential threat to biodiversity. We find that species with a preference for high-density vegetation and with shallower beak depth are most likely to be exposed to high severity fire, indicating a potential for habitat losses for species with these traits. Our findings reinforce calls for targeted management to reduce impacts of future fire where it is predicted to be outside the historical range of variation.

Global biodiversity patterns are the result of biogeographic mechanisms interacting with Earth systems processes over long time periods. Fire is one of the most pervasive biogeographic process that shapes ecosystem structure beyond climate and other abiotic factors[1,2] and plays a fundamental role in the global distribution of biomes[3]. Fire regimes (i.e., the characteristic fire intensity, frequency, and seasonality of a region) drive plant and animal biodiversity across the globe[4,5], promote diversification of adaptive responses, and facilitate coexistence of diverse species assemblages[6,7].

We are now seeing unprecedent changes in global fire activity with unknown implications for biodiversity[8]. In addition to direct human influences, changes in fire regimes interact with other global change factors like climate change, vegetation change, invasive species, and land use to influence biodiversity[1,9,10]. As fires outside the historical range of variation become more common[11], species are increasingly exposed to conditions outside their adaptive window[12], and for which their functional traits may be ill-equipped[13–15]. While changing fire regimes likely pose a threat to some species[16,17], other evidence suggests species' responses to novel fire characteristics may vary substantially, with both positive and negative responses being amplified[18]. With potentially significant implications for conservation, moving from retroactive assessment of fire impacts to proactively identifying regions most strongly exposed to changing fire regimes will be critical for understanding how changing pyrogeography will impact biodiversity.

Until recently, our understanding of the relationship between changing fire regimes and biodiversity at biogeographic scales has been limited by data availability. In this study, we leverage significant developments in both fire forecasting and species distribution modeling to investigate where future high severity fires might occur in the

[1]USDA Forest Service Rocky Mountain Research Station, Albuquerque, NM, USA. [2]Center for Fire Resilient Ecosystems and Society, University of New Mexico, Albuquerque, NM, USA. [3]Cornell Lab of Ornithology, Cornell University, Ithaca, NY, USA. [4]USDA Forest Service Aldo Leopold Wilderness Research Institute, Missoula, MT, USA. [5]Present address: Ariel Re UK Limited, London, UK. ✉e-mail: kari.norman@usda.gov

western United States, and which species will be most exposed when they burn. We develop comprehensive maps of species richness, community uniqueness, and functional diversity of bird communities and assess predicted exposure to low or high severity fire across these three facets of biodiversity. We use birds as our focal taxa due to comprehensive data availability for species in time and space and a strong history of birds as indicators[19]. We focus on the western United States as one of the most fire-prone regions on earth[20], containing multiple global biodiversity hotspots[21,22], and showing substantial evidence of rapidly changing fire regimes[23,24].

To evaluate the potential exposure of bird communities in the western United States to high severity fire, we first develop avian biodiversity maps using data from the eBird Status and Trends Project[25]. eBird Status and Trends leverages volunteer-collected bird survey data and rigorous spatiotemporal modeling to predict the relative abundance and population trends of birds at high spatial resolution while accounting for observer effects and other biases associated with participatory science data[26,27]. Using relative abundance maps for birds in the western United States, we construct maps for three common biodiversity metrics: 1) species richness, or the count of the number of species in a given locale, 2) local contributions to beta diversity, or the relative uniqueness of a given locale in a landscape, hereafter referred to as "uniqueness"[28], and 3) functional diversity, which describes the trait space occupied by a given community. To calculate functional diversity, we pair all species in our study with species-level traits from the AVONET trait database describing morphology, habitat, and life history characteristics[29]. To assess predicted fire, we draw from recent efforts by Parks et al.[30], who developed 30 m resolution predictions of fire severity for all forested ecoregions in the western United States with a measurable fire regime[31]. Predictions are based on the climate and landscape conditions at the time the product was developed (2016) and assign high severity (i.e. stand-replacing in this system) or low severity fire as more likely if a fire were to burn today. Pairing avian biodiversity maps with predicted fire severity resampled to a 3 km resolution, we identify hotspots of avian biodiversity and classify them as "areas of concern" (predicted high severity fire), "refugia" (predicted low severity fire), or "mixed" (similar levels of predicted fire severity classes). These designations reflect strong evidence of low severity fire as a resilience mechanism and high severity fire as a destabilizing mechanism in these systems[32,33]. We further identified individual species for whom a significant portion of their global population is predicted to be exposed to high severity fire and model the relationship between exposure level and species traits. By leveraging these diverse datasets, we provide an unparalleled look at the biogeography of not just altered fire

regimes, but multiple facets of avian biodiversity inhabiting areas predicted to experience stand-replacing fire.

## Results

Biodiversity maps showed high variation across space and the three biodiversity metrics (Fig. 1). We found that up to 12% of the study area contained both high levels of biodiversity (top 30% of the distribution for a given metric) and occurred in areas predicted to burn at high fire severity (species richness 12%, uniqueness 11%, functional richness 8%; Fig. 2a). Up to 27% of the study area had high biodiversity in areas predicted to burn at low severity (species richness 27%, uniqueness 18%, functional richness 20%). We identified biodiversity hotspots at the HUC12 watershed scale, the smallest hydrologic unit defining a basin's drainage that is also a common unit for management assessments. The 10% most biodiverse watersheds for each ecoregion were considered hotspots, a threshold that allows for sufficient spatial aggregation for regional interpretation and accounts for range-restricted species more effectively than more restrictive cutoffs[22,34]. Over half (55–58%) of hotspots were classified as refugia, where high biodiversity intersected areas with predicted low severity fire. However, 24–30% of hotspots were classified as areas of concern, where high biodiversity intersected areas with expected high severity fire (Fig. 2b). Within these areas of concern, 52–60% occurred in areas with historical low-severity fire regimes, indicating a fire regime mismatch and a potential biodiversity threat (Fig. 3).

To assess if biodiversity hotspots were disproportionately found in predicted low or high severity areas relative to the background landscape, we performed left- and right-tailed binomial tests parameterized by the observed ratio of low to high severity fire in each ecoregion at the pixel scale. Across ecoregions and metrics, biodiversity hotspots were more often found in areas expected to burn at low severity than would be expected based on the distribution of fire severity for an ecoregion (Fig. 4). Only the Great Basin ecoregion showed biodiversity hotspots predicted to occur significantly more frequently in high severity areas for all three biodiversity metrics. For some ecoregions, whether higher biodiversity values were found more frequently in areas expected to burn at high rather than low severity depended on the biodiversity metric examined (Fig. 5, Supplementary Fig. S1). Ecoregions generally fell into two groups, 1) high congruence between biodiversity distributions for high and low severity areas (e.g. West Cascades), and 2) distributions differing in central tendency (e.g. East Cascades).

The distribution of percentage of total global population for each species exposed to predicted high severity fire had a long right tail, with five species standing out as having a substantial portion of one of

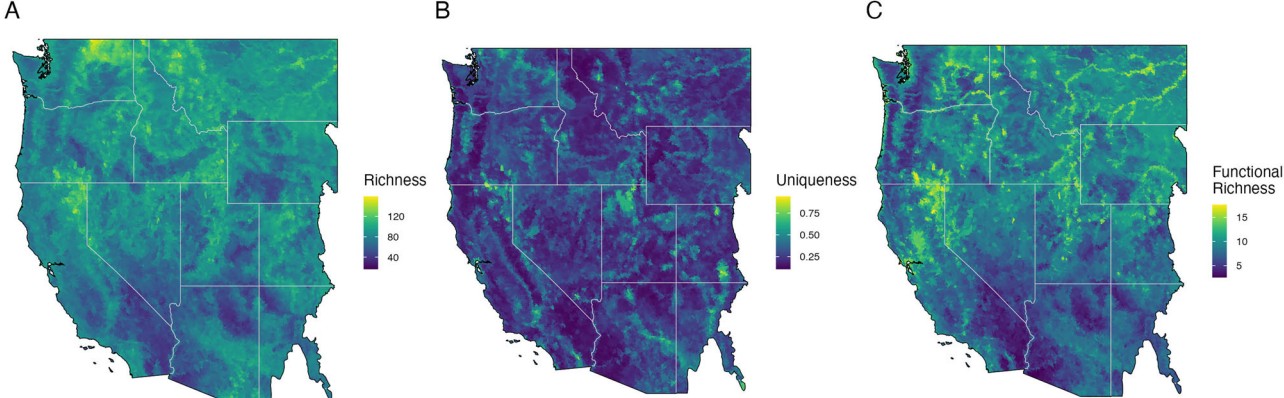

**Fig. 1 | Biodiversity metric maps.** Maps species richness (**A**), local contributions to beta diversity (**B**), and functional diversity (**C**) for our study region, inclusive of all the bird species in the western United States. Metrics are summarized as the mean value for each HUC12 watershed. Basemaps were made with Natural Earth. Free vector and raster map data @ naturalearthdata.com.

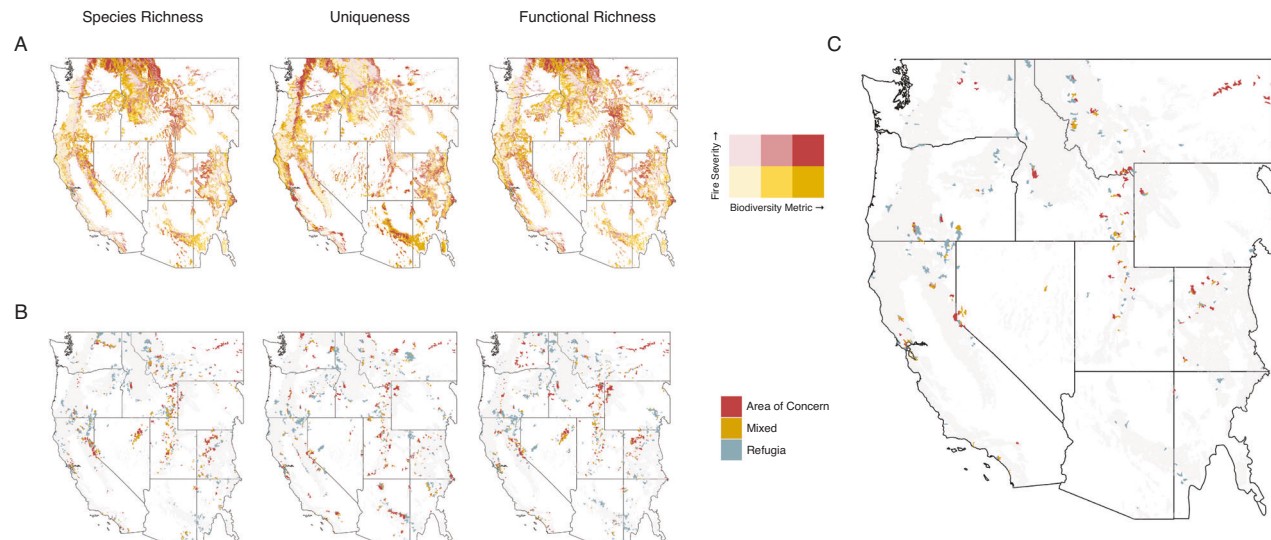

**Fig. 2 | Maps comparing biodiversity distributions to fire severity including identified hotspots. A** Shows bivariate maps of biodiversity metrics by predicted fire severity, with higher values of biodiversity in darker values, low severity fire in yellow, and high severity fire in red. **B** Shows the hotspots for those metrics broken down by type, with "areas of concern" shown in red, "mixed" shown in yellow, and "refugia" shown in blue. Maps visualize species richness (column 1), local contributions to beta diversity (column 2), and functional richness (column 3). **C** Shows hotspot areas congruent across all three biodiversity metrics, also broken down by type. Hotspots are identified at the HUC12 watershed level. The light grey footprint indicates the forested study area extent. Basemaps were made with Natural Earth. Free vector and raster map data @ naturalearthdata.com.

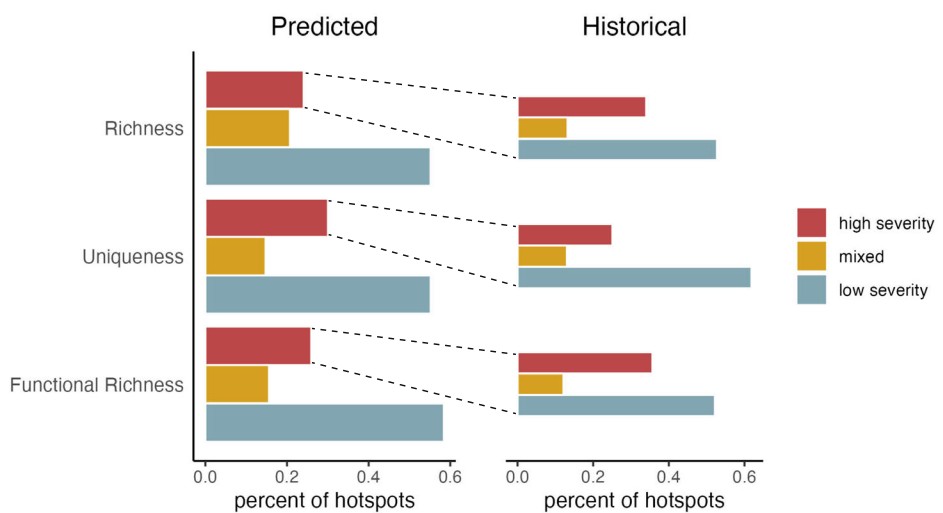

**Fig. 3 | Percentages of current and historical hotspots in each hotspot type.** Percent of hotspots for each biodiversity metric distributed across fire types, with predicted future fire severity shown in the left panel, and fire severity of the historical regime for only the predicted high severity hotspots given on the right. Source data are provided as a Source Data file.

their seasonal populations exposed including Mountain Quail (34% of resident population), Brown-capped Rosy-finch (36% of nonbreeding population), Williamson's Sapsucker (36% of breeding population), White-headed Woodpecker (39% of resident population), and Flammulated Owl (59% of breeding population) (Fig. 6). For insight into the characteristics of species disproportionately exposed to high severity fire, and potential areas of community-level vulnerability, we identified the traits related to the percentage of a species' population in the forested study area found in expected high severity areas. We performed beta regressions with species traits as predictors and the population percent in high severity as the response variable for both breeding and nonbreeding populations. We found that birds with shallower beak depth (e.g. Calliope Hummingbird, Golden-crowned Kinglet) and a preference for densely vegetated habitats (e.g. Brown Creeper, Pacific Wren) were significantly more likely to be exposed to

predicted high severity fire. Habitat density was not a significant predictor for nonbreeding communities, though beak depth remained (see supplement for full model fits).

## Discussion

We showed macroecological-scale exposure of bird diversity to expected fire regime changes. Up to 30% of forest bird diversity hotspots in the western United States are predicted to be exposed to future stand-replacing high severity fire, which could remove habitats for many forest-dependent species in the short and potentially long-term. However, up to 58% of bird diversity hotspots occurred in areas with predicted low severity fire, highlighting opportunities to identify and actively manage potential biodiversity refugia[35,36]. As species responses to high severity fire are highly diverse, area of concern hotspots notably do not represent vulnerability, but rather areas where

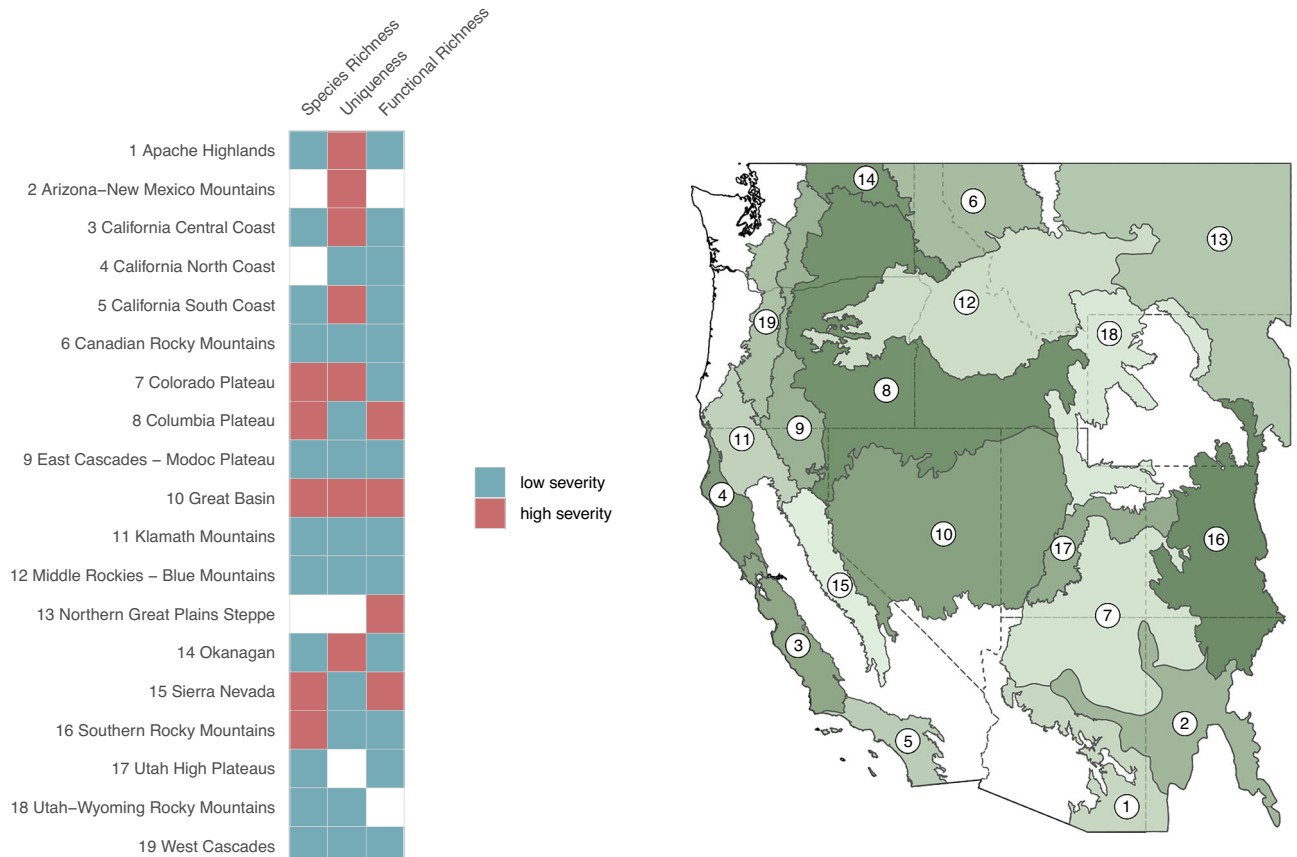

**Fig. 4 | Comparison of hotspot distribution across severity types to background ecoregion distributions.** Results of the lower and upper tail binomial tests comparing the ratio of hotspots in low and high severity fire to the background landscape ratio. Ecoregions that had a significant test for a given biodiversity metric are shaded, with blue indicating hotspots falling significantly more frequently in lower severity areas than expected based on the background landscape ratio and red indicating higher severity. The ecoregion number references the ecoregion map on the right. The ecoregion basemap was made with Natural Earth. Free vector and raster map data @ naturalearthdata.com. Source data are provided as a Source Data file.

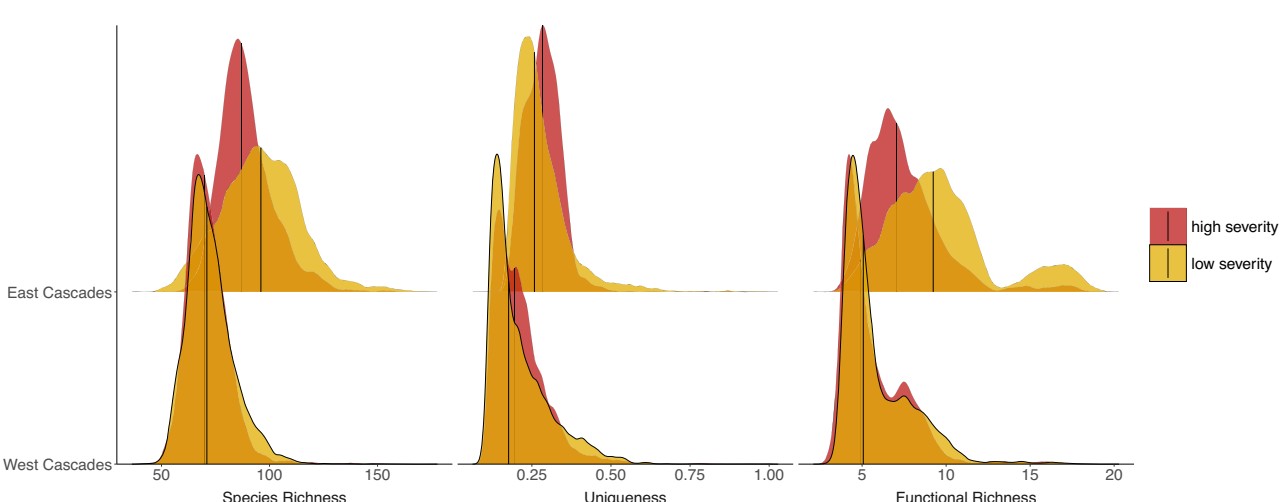

**Fig. 5 | Density plots of biodiversity metric distributions across fire severities.** Bivariate density plots showing the distribution of biodiversity metrics across low severity (yellow) and high severity (red) areas for two representative ecoregions, with vertical lines indicating the distribution median. Ecoregions generally fall in two categories: high congruence in how biodiversity is distributed across the predicted severity types (West Cascades) and mismatch in mean tendency, variance, or kurtosis (East Cascades). See Supplementary Fig S1 for all ecoregions included in the study.

biodiverse communities face greatest exposure. In some areas, there is a substantial mismatch between the fire regimes historically experienced by bird communities and the predicted fire regimes, indicating potential for significant negative impacts on the bird communities in those areas (Fig. 7). While our study focuses on forests of the western United States, our approach is well suited for adaptation to the many regions of the world experiencing fire regime change (e.g. the Mediterranean[37], Australia[18], Boreal forests[38], Tropical forests[39]),

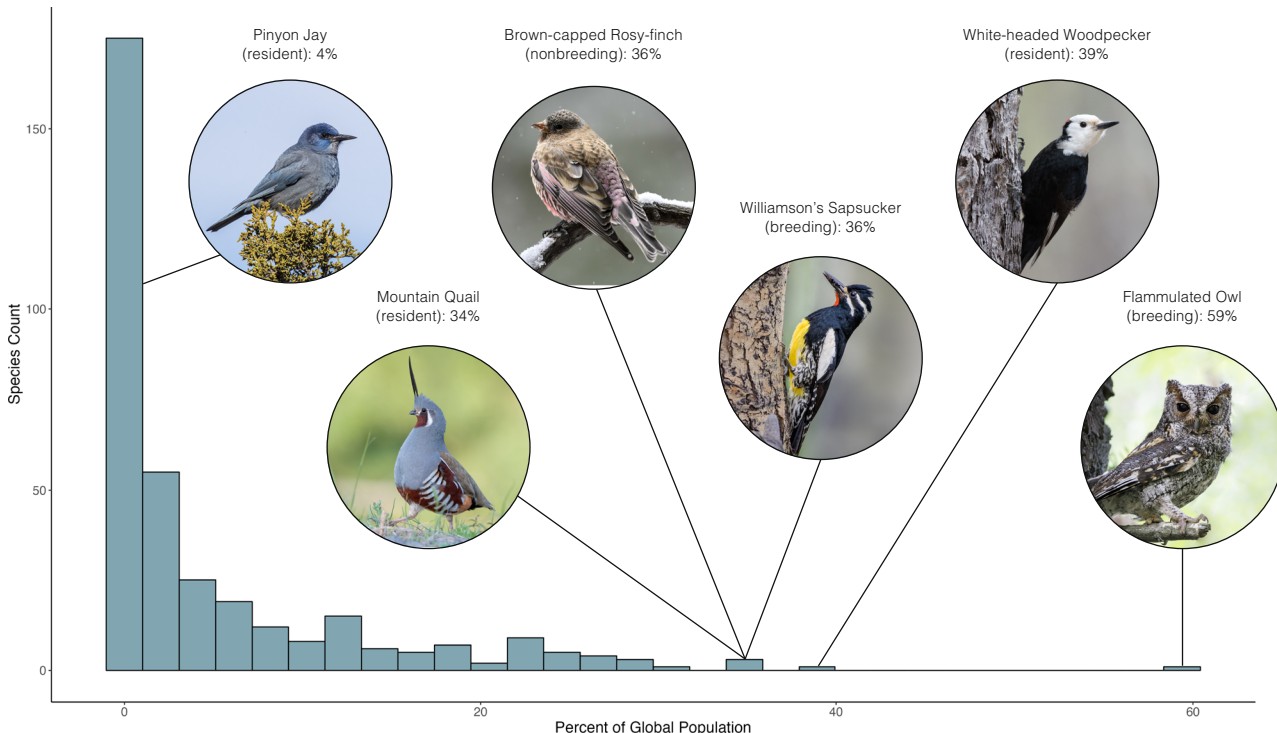

**Fig. 6 | Histogram of the number of species at different levels of population exposure to high severity fire.** Histogram showing the number of species for a given percentage of global population exposed to high severity fire, with breeding and nonbreeding ranges treated separately. The population percentage for the top five most exposed species are illustrated on the graph, as well as a representative low-exposure species. Species with less than 10% of their total population in the forested study region are excluded. Photos sourced from the Cornell Lab of Ornithology | Macaulay Library: Pinyon Jay (ML310160781), Mountain Quail (ML586116811), Brown-capped Rosy-finch (ML626811381), Williamson's Sapsucker (ML247159221), White-headed Woodpecker (ML86731361), Flammulated Owl (ML580715921). Source data are provided as a Source Data file.

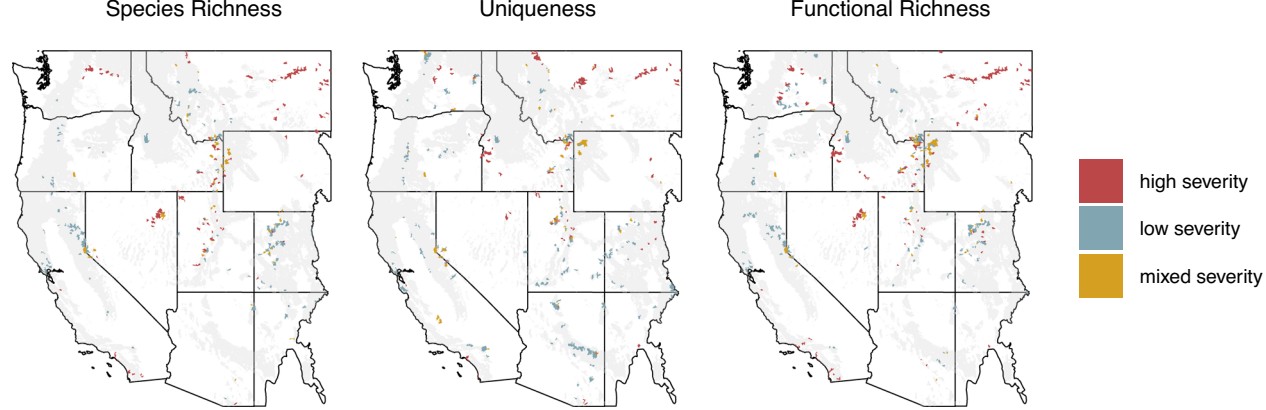

**Fig. 7 | Maps of area of concern hotspots broken down by historical fire regime.** Maps of the historical fire regime for only "area of concern" biodiversity hotspots, or those predicted to experience high severity fire. The light grey footprint indicates the forested study area extent. Hotspots in these maps are also reflected in histogram form in Fig. 3. Basemaps were made with Natural Earth. Free vector and raster map data @ naturalearthdata.com.

requiring only robust predictions of fire severity and species distribution maps for the region and taxa of interest.

Our assessment points to regions predicted to have particularly high or low exposure to high severity fire. Encouragingly, our analysis showed that most regions had more hotspots classified as refugia than expected based on the ecoregion's background distribution of fire severities (Fig. 4). Uniqueness hotspots were most likely to be disproportionately exposed to high severity fire, indicating that compositionally distinct communities might be at elevated exposure. Hotspots for all three biodiversity metrics in the Great Basin were predicted to be more heavily exposed to high severity fire than expected, alongside the Columbia Plateau, Colorado Plateau, and the Sierra Nevada, each of which had two disproportionately exposed metrics. These regions, with the exception of the Sierra Nevada, historically experienced predominantly high severity fire regimes, indicating that high biodiversity areas may be supported by the predicted exposure to high severity fire. At the scale of the entire western United States, regions with multiple areas of concern include geographies with well-documented changing fire regimes, including Lake Tahoe, the Greater Yellowstone Ecosystem, and the Colorado Rockies[40,41].

The implications of high severity fire for avian biodiversity are most concerning where high severity fires represent a significant

departure from historical fire regimes, as species may be poorly adapted to relatively novel stand-replacing conditions. Comparing area of concern hotspots to the LandFire historical fire regimes data product[42], we found that the majority of biodiversity hotspots occurring in predicted high severity areas historically experienced low-severity fire regimes (Fig. 3). Multiple regions stand out as being strongly mismatched with predicted high severity fire (Fig. 7), including hotspots across the Sierra Nevada and Cascades, the Bay Area of California, the Mogollon Rim, the Colorado Rockies, and forested areas in Utah. Conversely, biodiversity hotspots in central and eastern Montana, eastern Washington, and Nevada historically experienced high severity regimes, suggesting that the predicted high severity fire in those regions could act to maintain biodiversity.

Of greatest concern for the persistence of avian forest communities is not just exposure to high severity fire, but the potential for stand-replacing fire to initiate an ecosystem type conversion to non-forest habitat types. Conversions catalyzed by fire occur when vegetation is unable to regenerate after burning due to frequent reburning[44], limitations in wind-driven seed dispersal due to large patches of stand-replacing fire[45], and inhospitable environmental conditions for seedling establishment due to climate change[46]. While some generalist bird species may be able to persist in the novel vegetation conditions, type conversion is a significant threat to forest-specialists and can lead to loss of avian species richness in general[43]. The magnitude of species loss or turnover is likely dependent on how diverged the new habitat type is from pre-fire conditions and how specialized the pre-fire community was on the original habitat type. Recent estimates suggest that in the western United States up to 50% of the landscape is at risk of experiencing type conversion in the coming decades[47]. Type conversion may be particularly likely in parts of the southwestern United States and for areas like the Kaibab Plateau where our analysis found hotspots of uniqueness, many of which are predicted to burn at high severity[48,49]. Likewise, up to one-third of the Klamath region around the Oregon-California border is predicted to convert to shrub or hardwood away from conifer-dominated forest types[50], an area we found to have a substantial number of hotspots for all three biodiversity metrics. These examples of areas where predicted high severity fire and high avian diversity align with high type conversion risk illustrate a need for broad-scale spatial assessments of the relative risk of type conversion across the western United States to appropriately prioritize management efforts.

Assessment of the functional traits most strongly related to high severity exposure across a species' population indicate that species with a preference for high density habitat will be most highly exposed to high severity fire. This finding has profound implications for the persistence of these species as dense forests species, often found in fire-suppressed mature and old-growth forests, have already lost significant habitat to stand-replacing fire[51]. The most highly exposed species are in this case also the most likely to experience negative impacts from a high severity fire event. In addition, we found that birds with shallower beaks tended to have higher exposure, which includes species like hummingbirds and flycatchers. With a less obvious relationship between trait and fire severity, further species-level investigations will be necessary to assess individual species' response to fire severity. These results illustrate the strength of a trait-based approach for identifying groups of species of management concern due to disturbance, particularly when there may be a dearth of species-specific work. It also offers the potential to design management action based on groups that are likely to respond similarly to intervention due to their unifying characteristics[52].

While to our knowledge this is the most geographically comprehensive analysis of predicted avian biodiversity exposure to fire severity, inclusion of further realism in future analyses would allow for more precise predictions of regions of exposure. As our study assesses the likely fire severity *given that a fire occurs*, our study does not incorporate the factors influencing the probability of a fire occurring. Additional broad-scale work on spatial patterns of burn, continuity in fuels across landscapes, and ignition risks will be critical for assessing burn probability. Fire predictions are also based on mean fire weather conditions despite a fair amount of variability in the conditions compatible with fire, adding some degree of uncertainty to our results, though overall findings are not likely to be substantially impacted. Additionally, our study assesses only the fires burning in forests despite a considerable percentage of the area burned in the western United States being shrub and grassland ecosystems[53], which are themselves home to diverse bird communities and highly prone to type conversion[54–56]. Changes in fire regime look substantially different in these ecosystems making the forest fire severity-centric approach we take here less relevant, but similar analyses using metrics of change more applicable to those ecosystems will be critical for getting a complete view of fire impacts in the western United States[57].

Our study identifies specific geographies where development of a stronger understanding of how high severity fire will affect species and communities will be important. Biodiversity hotspots that are expected to burn at high severity, especially those with historical low severity fire regimes or at high risk of type conversion, should be prioritized for fuels reduction and forest restoration activities like thinning and prescribed burns to mitigate fire impacts[58–61]. This will be particularly critical for the highly exposed species with a preference for high density forests, which can benefit from management activities that restore fire as a habitat-generating rather than a habitat-reducing process[62,63]. While treatment at the watershed-scale identified here may be too large to be feasible in some areas, even smaller restoration efforts can maintain unburned patches that act as refugia to support future biodiversity regeneration[36]. At the individual species level, our results also revealed species with high exposure to predicted high severity fire, including the Flammulated Owl and Brown-capped Rosy-finch that, according to limited previous research, may not be well adapted to high severity fire[64]. The Brown-capped Rosy-finch stands out particularly as both endangered and severely range limited, therefore having the potential to be significantly impacted by even one or few large, high severity events. However, exposure to high severity fire is not always a conservation threat, and fire impacts to species are dependent on the fire's configuration and return interval, and a species' natural history[65–67]. The three other species predicted to be most highly exposed to high severity fire (White-headed Woodpecker, Williamson's Sapsucker, and Mountain Quail) are all classically fire-adapted species, whose predicted overlap with high severity fire may reflect a regime preference. We make the complete ranking of species available in the supplement as a potential tool for pairing exposure to existing natural history knowledge, and to further identify highly exposed species whose relationship to fire may be poorly understood.

## Methods

### Study area and data

Our study area is restricted to forested ecoregions in the contiguous western United States[68]. Ecoregions with low forest fire activity are excluded due to data limitations (e.g. Pacific Northwest Forests and Sonoran Desert), as described by Parks et al.[30]. To assess breeding and nonbreeding bird communities, we accessed the eBird Status data product from 2022 using the *ebirdst* package (data version 2021 and 2022; v3.2022)[69] in R (v4.3.3, R Core Team 2024). Since not all species were available for the 2022 version we used the 2021 version to fill species gaps, with species occurrence data and percent of population estimates both robust to slight methodology changes between data versions. Models were based on checklist data collected starting on the first day of 2008 and ending on the final day of the product year (e.g. December 21 2022 for 2022 species). We obtained data for 2440 individual species, 571 of which occurred in our study area and were therefore included in the analysis. Maps were available for all species

occurring in the study area. The eBird Status data product includes maps of relative abundance at 3 km resolution across the range of each bird species generated from volunteer bird surveys collected in a semi-structured fashion, where participants record ancillary information on observation effort[70]. The modeling workflow uses systematically filtered checklist data in an adaptive spatiotemporal modeling framework[71,72] that incorporates observation effort, detectability, and remotely sensed environmental covariates to predict relative abundance at 3 km × 3 km for breeding, nonbreeding, and pre- and post-migration seasons[26,73]. In this framework, relative abundance is defined as the expected count of individuals by an expert birder during a 1-h, 2-km survey within the ideal time of day, weather conditions, and observer effort for detection of that species. The data version used for each species is listed in the supplement.

For functional diversity metrics we matched species to traits from the AVONET trait database[29] which includes six ecological variables (*habitat type, habitat density, migration, tropic level, trophic niche, primary lifestyle*) and 11 morphological traits (*culmen beak length, nares beak length, beak width, beak depth, tarus length, wing length, Kipp's distance, secondary 1, hand wing index, tail length, mass*). Together these traits describe the functional dimensions governing a species' ecological role as completely as possible for all birds included in our study. For birds, like most animal groups, general traits related to fire vulnerability or response have not been well identified, and we therefore cast a broad net to identify general associations[74,75].

Fire severity data were obtained from Parks et al.[30], which gives predicted fire severity should a fire occur in the 19 western US ecoregions where fire is prevalent. Predictions are based on variables that summarize live fuel, topography, climate, and fire weather for conditions in 2016. We do not expect that these variables have changed significantly since being measured with the exception of live fuel, which is impacted by fires that have burned in the interim period. Rather than invalidating the hotspot maps, these discrepancy areas represent realizations of the refugia or area of concern designations. For our application, high severity fire is defined as ≥95% canopy mortality, and is therefore equivalent to the qualitative designation of stand-replacing fire[76]. The original raster datasets (from Parks et al.[30]) depict the probability of high severity fire, were a fire to occur, under the mean weather conditions for which fires occur, with a spatial resolution of 30 m. We resampled each ecoregional dataset to match the resolution of the eBird datasets (3 km). Because there were differing proportions of high severity fire (vs. moderate/low severity fire) within each ecoregional model in Parks et al., the probability of high severity fire is not comparable among ecoregions. Accordingly, we used ecoregion-specific probability thresholds to translate probability of high severity fire to a binary high and moderate/low severity map to allow for comparison across ecoregions following Parks et al.[49] and Davis et al.[31]. The thresholds were determined to ensure that the proportion of high severity fire in the 3 km binary raster datasets were identical to the proportion of high severity fire in the input datasets used in the original Parks et al. models. In the Sierra Nevada ecoregion, for example, 32% of pixels in the original dataset were classified as high severity fire. We wanted to ensure that 32% of pixels in the rescaled, 3 km binary dataset were also classified as high severity, so we used the 32% probability threshold to classify the 3 km dataset into high vs. moderate/low severity.

## Analysis

We constructed 3 km resolution rasters of community-level biodiversity metrics based on the eBird species relative abundance rasters for breeding (*n* = 520 total species) and nonbreeding (*n* = 502 total species) communities, with both including resident species. Since relative abundance values are not comparable across species, abundance estimates were converted to binary occurrence values for community-level metrics. Occurrence was inferred as any non-zero

pixel (sensu Ng et al.[77]) and we calculated pixel-level biodiversity metrics using the 3 km pixel as an individual site. Using these occurrence layers as a starting point, we generated maps for three separate biodiversity metrics: 1) species richness, 2) uniqueness (i.e., local contributions to beta diversity), and 3) functional diversity. First, we calculated species richness as the sum of all binary occurrences in each pixel across the study area. Next, we calculated the relative uniqueness independently for each ecoregion to account for ecoregions with inherently higher or lower beta diversity. This metric was implemented in the language Julia to make calculations computationally feasible across millions of cells. Third, we calculated functional diversity, which describes the volume of the functional space, for each 3 km pixel using the *fd_fric* function from the *fundiversity* R package (v1.1.1)[78]. We included all AVONET traits and prepared them for analysis by: 1) z-score scaling continuous traits, 2) computing functional dissimilarity based on Gower's distance, which allows for categorical traits, using the R function *gower.dist* from the package *StatMatch* (v1.4.2)[79], and 3) performing a principal components analysis on the dissimilarity matrices using the function *dudi.pco* from the package *ade4* (v1.7-220)[80]. We used the first four PCA axes to describe the functional space, following best practices for balancing sufficient information and computational feasibility[81]. We also calculated functional evenness (function *fd_feve*) and functional divergence (function *fd_fdiv*), and all metrics for nonbreeding communities, which are visualized in the supplement.

To identify geographic areas of particular concern, we summarized mean values of the biodiversity metrics in forested areas at the scale of a HUC12 watershed. We identified hotspots as the watersheds with the top 10% highest values for a given metric. We then classified hotspots as "areas of concern" if they contained more high severity fire than low, "refugia" if they contained more low severity fire than high, and "mixed" if the percentage of low and high severity pixels were within 10% of each other. Since there is significant variation in the baseline values for metrics across ecoregions (e.g. coastal ecoregions have higher richness than desert ecoregions), hotspots were identified for each ecoregion independently. All visualizations of biodiversity metrics and hotspot locations used basemaps obtained using the R package *rnaturalearth (v1.0.1)*[82].

To assess how the most biodiverse areas were distributed across predicted fire severities, we took the top 10% of forested biodiversity pixels for each ecoregion and metric and classified them as high or low severity fire areas. For each ecoregion, we then performed binomial tests parameterized by the ecoregion-level ratio of total high severity vs low severity pixels test for statistical differences between predicted fire severities in biodiversity hotspots compared to all pixels in the ecoregion. We performed both upper- and lower- tail test to identify ecoregions where high diversity was more often in high severity than expected by chance or more often in low severity.

To assess the exposure of individual species to potential high severity fire, we used relative abundance maps from eBird Status to calculate two different population percentage metrics. First, the percent of the total global population found in predicted high severity fire in our study area, second, the percent of the forested population in the study area in high severity areas. For example, a value of 10% for the forested population percentage would indicate that, relative to the total number of individuals of the species within the study area (forested areas in the western US), 10% of these individuals overlap areas with expected high severity fire. We visualized the first metric, percent of the global population in predicted high severity fire, for all species with >10% of their population in the forested study area (Fig. 6) to identify species of particularly high exposure to predicted high severity in our study area. Some species in our study may be highly exposed to other kinds of changing fire regimes elsewhere in their range, which is not accounted for here. We then identified species characteristics most closely associated with high severity fire exposure

using beta regressions with the second metric, percent of forested population in predicted high severity areas, as the response variable and the original untransformed AVONET traits as predictor variables. Beta regression assumes that the data-generating process follows a beta probability distribution and is commonly used for modeling percentages[83,84]. We excluded species with less than 10% of their total population in western US forest to exclude species with very little of their range in the study area and ensure models were assessing differences within forest-dwelling birds rather than distinguishing between forest and non-forest dominant species. We split traits into three categories: morphology, habitat, and lifestyle, and performed a beta regression for each category separately for breeding and non-breeding communities (see supplement for model parameterizations). To account for correlation between variables, 3 morphological traits (*nares beak length, beak width, wing length*) and the trophic level variable, which was a higher-level descriptor of trophic niche, were not included as predictor variables.

In order to compare historical fire regimes to predicted regimes, we summarized the LandFire historical fire regimes data product at the HUC12 watershed level[42]. As this data product also separates regimes by return interval, we considered any return interval characterized by high severity as historically high severity and low and mixed severity regimes of any return interval as low severity. Watersheds with majority historically high or low severity pixels were classified as such, with mixed severity identified if the percentage of historically low and high severity pixels were within 10% of each other. Since the LandFire data product uses a ≥75% canopy mortality definition of high severity, whereas our data product uses a more common ≥95% canopy mortality cut off, our direct comparison likely overestimates the area that was historically high severity following our severity definition. Our results are therefore a conservative estimate of the number of hotspots experiencing a regime mismatch.

### Reporting summary

Further information on research design is available in the Nature Portfolio Reporting Summary linked to this article.

## Data availability

eBird relative abundance datasets are available online at https://science.ebird.org/en/use-ebird-data/download-ebird-data-products or through the R package or public API. The predicted fire severity data product is available for download through the Fire Research and Management Exchange System (FRAMES; www.frames.gov/NextGen-FireSeverity). The AVONET trait database is archived at https://figshare.com/s/b990722d72a26b5bfead. Rasters of biodiversity metrics and other interim data products are archived for public download in the Zenodo repository https://doi.org/10.5281/zenodo.15414728[85]. Source data are provided with this paper.

## Code availability

Code associated with the analyses are archived in the Zenodo repository https://doi.org/10.5281/zenodo.15413872[86].

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

## Acknowledgements

We wish to thank Shirley and Allan Casey for support, and Dr. Sarah Sawyer for valuable comments on early versions of the manuscript. We also thank the many participants of the eBird data collection process, developers of included data products, and the image contributors to the Cornell Lab of Ornithology | Macaulay Library.

## Author contributions

K.E.N. contributed to conceptualization, data curation, formal analysis, and writing; A.N.S. contributed to conceptualization and writing; S.A.P. contributed to conceptualization, data curation and writing; C.L.D. contributed to conceptualization and writing; and G.M.J. contributed to conceptualization and writing.

## Competing interests

The authors declare no competing interests.
