## [Transparent Peer Review file · Nature Communications]

Exposure of western United States bird communities to predicted high severity fire

Corresponding Author: Dr Kari Norman

Version 0:

Reviewer comments:

Reviewer #1

(Remarks to the Author)

Thanks for the opportunity to review the manuscript "Exposure of western United States bird communities to predicted high severity fire." The manuscript presents the results of a novel approach combining species distribution models and fire forecasting to assess the exposure of bird species to future fire regimes in western United States, identifying both areas and species of conservation and management concern.

The manuscript is well written and communicated, the analyses overall well justified and the results noteworthy. However, I have a number of concerns specifically relating to clarity of methodological details and important context in which the results need to be placed.

First, the methods make clear that the change in fire regime at the focus of this study is places likely to experience severe fire, should a fire occur in a location. This is a very important nuance to the paper that needs to be made clear throughout the introduction. Further, from reading the methods of the study referred to as the source of the fire severity predictions, it's not clear how future fire severity is modelled given the Parks et al. 2018 study only provides predictions for the year 2016. The authors need to make clear whether they are using a different set of predictions from the Parks et al. model and describe the methods used to generate those (given the study has been described as giving predictions of future fire severity exposure, Line 58) or clarify the framing of the paper as being about 'current' fire severities if it's the 2016 predictions being used. This is a very important part that needs to be clarified in future versions of the manuscript as it impacts how the results can be framed.

Future fire modelling is inherently uncertain, however the results presented here appear to have used the mean predictions of the fire severity modelling despite the likely reasonably large uncertainties that drive fire severity (e.g. variation in daily fire weather & other future conditions). Given the main results are locations on maps, I think that there needs to be an acknowledgement of the uncertainties in the two major analyses of the manuscript (i.e. the hotspot analysis and the species exposure analysis). Given the lack of clarity in the manuscript about what conditions the predicted severities represent, it gets a little difficult to tell whether the authors are referring to historical or future severity values in different parts – need to make clear throughout (e.g. line 99).

Second, the manuscript focuses on the increase in bird species' exposure of high severity fire but does not consider the relative vulnerability of each species in the analysis to changes in the amount of high severity fire beyond the example species identified in the discussion. This is acknowledged throughout, but given the hotspot analyses used, it's highly plausible that the at risk hotspot locations would change if species' vulnerability to severe fire was considered. I think that the authors need to at least acknowledge this.

Lastly, I think the functional traits analysis, while informative, needs a stronger justification for inclusion in the study. Why was this analysis done and why were these specific traits chosen? Is there a particular hypothesis the authors have about these traits in terms of vulnerability to severe fire? At present it's a little unclear if these traits were chosen based on data availability, rather than because they represented the major functional differences between species relevant to fire-related analyses.

Line comments below

L50: Impacted? Need to replace with exposed as have not considered variability in vulnerability to high severity fire.

L58: Importance nuance here – this paper doesn't assess species exposure to future fire regimes, it assesses species exposure to high severity fire. Make sure this is consistent throughout.

L63: Occurrence here, but relative abundance elsewhere (e.g. L237). I can see that different metrics are used at different points, but I think this needs to be made clearer through the manuscript. Especially when talking about the study's broader results.

L68-69: Not clear in this sentence how the trait database was used – might need a second sentence here saying what was done with the trait database to pre-empt the methods.

L71: Fire severity under what scenario of climate? At what time point in future?

L80: To a non-American reader, may need to make it clear here or earlier where severity is introduced whether all high severity fire in all ecosystems in this study (western United States) is stand replacing or whether this statement needs to be nuanced a bit (i.e. are there some ecosystems where severe fire is not stand replacing?).

L83: What is meant by top 30% of biodiversity distribution? Is this the top 30% of richness values? Or something else?

L86: State the range of percentages in parentheses here

L88: Why was 10% chosen as the threshold for a hotspot?

L104: I'm not quite clear on what is meant by "high and low severity biodiversity distributions" here – do you mean high biodiversity values affected by high and low severity fire or something else?

L107: Wording is a bit clunky here. Suggest including the word 'species' so it reads 'percentage of a species' total global population'

L112: I would like to see a stronger justification of the traits analysis. Why is it important to look at which traits are more likely to be exposed to high severity fire?

L123: How large an increase is this from the historical fire regime?

L127-129: It would be good to give an example or two here from the results to help crystallise it for the reader.

L152: I think it would be good to quantify this more formally. What is the proportional increase in high severity fire in these locations?

L185-186: This is key and I think could be drawn out a bit more in the manuscript. Is it possible that the hotspots could change locations if just focused on species particularly vulnerable to high severity fire, as distinct from the all species approach taken here.

L190: This study referred to here is not in the reference list and wasn't provided to reviewers.

L229-241: More detail needed here on which species were included. How many bird species have been modelled? What proportion of western United States bird species overall? Over what time period were records collected from? I know some of this is included in the eBird references, but it's important context for this study and so should be included.

L243-246: Why were these traits chosen? Was this just based on what data were available? Were there other traits likely to be important that weren't captured by these traits that may change the functional traits space developed in lines L270-275?

L246-255: I think that the fire severity predictions used in this study need further information. I understand that the authors reference the original study but there's important information in the original severity modelling studies that is important for understanding the present study here. For example, the abstract (e.g. L13) and introduction (e.g. L58) sets the paper up as being about assessing exposure to changing/future fire regimes, but it is not clear from the information presented here how the change from 'historical' fire regimes is being quantified. How does the Parks et al. studies predict future/changed fire severity? The methods in that study only describe making predictions to 2016 conditions. Has it since been used under some set of future conditions that are different from historical? Are these conditions in the near term, medium-term or long-term?

L249: The 95% canopy mortality = stand replacing statement needs to be supported with a reference.

L280: These thresholds need a stronger justification.

Figure 5: I think adding a dashed vertical line or similar to identify the mean/median of each density plot would be helpful, given the long tails of some of the distributions.

(Remarks on code availability)

Code was downloadable and included a README file. Appears to be enough information to re-run analysis in the Zenodo repository. I did not re-run the whole analysis though, I just viewed the code.

Reviewer #2

(Remarks to the Author)

Revision of the manuscript entitled “Exposure of western United States bird communities to predicted high severity fire” for Nature Communications

Thanks for the opportunity to review your manuscript. All of my comments and considerations below are intended solely to help improve the relevance of your manuscript, as I think your work is a great contribution to the fire ecology field.

First of all, I would like to congratulate you on the introduction, it is very well written, fluid, and well organized. Overall, your manuscript is of interest to wildfire specialists, ecologists, and others, even beyond your region. In this sense, however, this could also be a limitation, particularly in the introduction and discussion sections. From my point of view, it would be fantastic if these two sections were improved by incorporating a broader, more global perspective. Since understanding fire effects on biodiversity is a complex and widely studied challenge, I would encourage you to highlight how your work builds on or advances the recent research. To help on this, I suggest you consider referring to the following studies where appropriate:

1. Puig-Gironès, R., Palmero-Iniesta, M., Fernandes, P.M., Oliveras Menor, I., Ascoli, D., Kelly, L.T., Charles-Dominique, T., Regos, A., Harrison, S., Armenteras, D. and Brotons, L., et al., 2025. The use of fire to preserve biodiversity under novel fire regimes. *Philosophical Transactions B*, 380(1924), p.20230449. <http://www.doi.org/10.1098/rstb.2023.0449>
2. Kelly, L.T., Hoffmann, A.A., Nitschke, C.R., Pausas, J.G., Sanderfoot, O.V. and Tingley, M.W., 2025. Evolutionary Implications of Trait–Fire Mismatches for Animals. *Global Change Biology*, 31(7): p.e70368. <http://www.doi.org/10.1111/gcb.70368>
3. Puig-Girones, R., Brotons, L. and Pons, P., 2022. Aridity, fire severity and proximity of populations affect the temporal responses of open-habitat birds to wildfires. *Biological Conservation*, 272, p.109661. <https://doi.org/10.1016/j.biocon.2022.109661>
4. Arrogante-Funes, F., Aguado, I. and Chuvieco, E., 2024. Global impacts of fire regimes on wildland bird diversity. *Fire Ecology*, 20(1), p.25. <https://doi.org/10.1186/s42408-024-00259-x>

Secondly, I'm not comfortable evaluating your spatial modelisation, as it is outside my statistics expertise. Therefore, I have reviewed the section specifying the spatial model statistics, but I encourage the editors to find (if the other reviewers are also not familiar with this kind of spatial model) an expert in this specific spatial modelling approach to review that part. I know this may not be good news for the authors, but that is my honest opinion, and I prefer to be transparent about this important point.

Following are a few minor comments:

Line 67–68: Regarding the other two diversity metrics, could you provide more detail on what you mean by “a measure of the volume of the multi-dimensional trait space”?

Line 83 – 85: Split the sentence.

Line 88: Explain what “HUC12 watershed” means the first time that it appears in the manuscript.

Line 114: Use “Using these percentage” instead of “Using this percentage”.

Line 122 – 123: Although it is positive to focus on a specific region, this can reduce the perception of the global relevance of the work. At the same time, mentioning the applicability of the results to other parts of the world without providing examples limits the global applicability of the research. This is a pity. I know the focus is justifiable, but adding a comment about similarities with other regions could increase the impact. For example, there are other areas with similar types of ecosystems (Mediterranean regions and/or temperate forests) that could experience similar dynamics. In addition, a little more information could be added about the implications of the impact of these changes. For example, “destroying habitats” is a very general, sensationalist, and somewhat exaggerated statement (given all the knowledge we have about wildfire ecology). It would be useful to specify which habitats and why they are so critical to biodiversity.

Line 126 – 129: It would be useful to specify further how this exposure to low fire intensity may or may not benefit biodiversity. What types of refuges or protection mechanisms exist in these areas? How can this information help to design conservation strategies? I sincerely believe that you can get more out of this topic.

Line 133 – 137: Even with the reference to LandFire (2010), it would be useful to include more citations to recent or more detailed studies that reinforce the conclusions. It would also be good to compare it with data from regions outside the LandFire system to reinforce the conclusion. At the same time, the “mismatch” between historical and predicted regimes is important, but there isn't enough discussion about how this mismatch affects species or communities. Explaining how these discrepancies can have ecological, or conservation consequences would add some depth to the manuscript. Another consideration is that changes in fire regimes can quickly alter these perceptions. It would be good to highlight this through

the bibliography. And if possible, management strategies to avoid adverse effects are also provided, this would be of great interest and help to managers.

Line 158 – 160: The discussion on habitat conversion lacks critical reflection on the methodological limitations of addressing this phenomenon. For example, the model may not be able to fully predict the long-term effects of habitat conversion, or there may be uncertainties in associating the effects of fires with changes in ecosystem type.

Line 158 – 167: You need to better connect the implications for forest management and biodiversity conservation. How can the results be used to design specific management strategies for biodiversity conservation or post-fire restoration?

Line 164 – 166: The persistence of species (site tenacity) must be taken into account, which can last for one or two generations, a fact that could “limit” the loss of certain species in the short term. It would also be important to consider the role of fire refuges (unburned patches) within burned areas that minimize the loss of certain species. It would be interesting to mention the need to maintain these refuges for wildlife as a post-fire strategy, especially in areas where high-intensity fires are expected. See for inspiration: <https://doi.org/10.1016/j.foreco.2023.121439>, <https://doi.org/10.1111/j.1469-1795.2012.00542.x> and their refs.

Line 173 – 176: It would be really helpful to give more context on why this conversion is important for biodiversity. What other studies or models support this prediction? Which species are most affected by this conversion? Keep in mind that there are species (mostly from open habitats) that do well with fires and also contribute to biodiversity. At the same time, although there is a call for the need for large-scale assessments, some specific recommendations or guidelines could be provided for those responsible for forest management or biodiversity conservation. This would improve the usefulness of the study and its practical implications. See for inspiration <https://doi.org/10.1038/s43247-025-02165-9> and their refs.

Line 191 – 203: Good point, but some of my previous comments may help to improve these limitations and future section.

Line 208 – 210: Good point.

Line 224. How many 3 km x 3 km pixels you finally analysed after data curation?

Line 228: Put a comma after “communities”: “To assess breeding and nonbreeding bird communities, we [...]”.

Line 230 – 231: The inclusion of eBird’s data product from 2022, matched with bird trait data from AVONET, provides a strong foundation for modelling. However, missing species were accounted for using the 2021 version, which is reasonable given computational limitations, but mentioning how any discrepancies between the two datasets were handled would strengthen this section.

Line 250: “non-transferable” instead “nontransferable”?

Line 242 – 255: The explanation of how data was adapted to account for ecoregion-specific fire probabilities adds robustness. This ensures that analyses across regions are comparable. However, more details on why these thresholds were selected, or how variable they are across ecoregions, would be helpful.

Line 274: Put a comma after “space”: “We used the first four PCA axes to describe the functional space, following best practices [...]”.

Line 291 – 292: The use of upper- and lower-tail tests is a good approach to capture extremes in fire-severity impact. However, providing more details on the binomial tests’ parameters would make it easier for replicability transparency (e.g., how the expected proportions were set).

Line 288: Put a comma after “ecoregion”: “For each ecoregion, we then performed [...]”

Line 303 – 314: This approach is strong because it integrates ecological traits with exposure to high-severity fire, providing insights into which species are most vulnerable. The exclusion of non-forest species is reasonable, and the choice of traits (morphology, habitat, lifestyle) for regression modelling is appropriate. However, it would be helpful to provide more details on the beta regression models, such as the model assumptions and how AVONET traits were used as predictor variables.

Line 315 – 325: The cautious approach of treating the mismatch between historical and predicted fire regimes as conservative is good for transparency. However, the impact of using different severity thresholds on the comparison could be better articulated (i.e., how much of an overestimate is expected and why it matters). Moreover, more details on the specific methods used to compare these regimes (e.g., statistical tests or methods of comparison) would improve replicability.

(Remarks on code availability)

Version 1:

Reviewer comments:

Reviewer #1

(Remarks to the Author)

The authors have done a great job responding to the previous round of reviews and suggestions. They have addressed all my queries adequately and the paper is much improved.

I don't have anything more substantive to suggest or comment on with this latest version, beyond encouraging the authors be careful of the language in the manuscript regarding "changing fire regimes" given the additional detail provided about the fire severity model being from 2016 (so the paper is measuring bird species exposure to past changes in potential fire severities), rather than explicitly estimating future severities. 'Altered' or 'Changed' fire regimes may be more appropriate as those terms have a past tense and is a clearer reflection of the analyses actually done in this paper.

(Remarks on code availability)

I reviewed the code in the first round of reviews.

Reviewer #2

(Remarks to the Author)

Revision of the manuscript entitled "Exposure of western United States bird communities to predicted high severity fire" for Nature Communications

Thanks for the re-opportunity to review your manuscript. Again, my comments and considerations are intended solely to help improve the relevance of your manuscript.

The authors have satisfactorily addressed most of the questions and comments raised by the reviewers, tackling methodological concerns, clarifying key points, and incorporating suggestions to enhance the clarity and relevance of the manuscript. In this regard, you have done a good job, and overall, I am satisfied with the revised document.

You have made significant adjustments to the text, including more detailed explanations of the data used, the limitations of the models, and the rationale behind the functional trait analyses. You have also expanded the discussion on the global implications of the study and incorporated relevant references suggested by the reviewers.

However, there are a few points that could benefit from further elaboration (apologies if I reiterate aspects from the initial review, we all have our biases and areas of expertise), and I encourage the authors to develop these more thoroughly in the text:

- Impact of habitat conversion. While the authors speculate on regions vulnerable to habitat conversion, they could delve deeper into the specific ecological implications of this phenomenon, including examples of how it affects forest and generalist species.
- Post-fire refugia. The authors mention that they are unable to address landscape-level burn patterns, but a more detailed discussion on the importance of unburnt refugia within fire-affected areas would be valuable. I encourage further development of this section, which I believe is fundamental and highly relevant to fire ecology and biodiversity.
- Global applicability. Although examples of similar systems outside the study region have been added, the discussion could be expanded further to explore how the findings might be useful for other regions with comparable fire regimes, such as Mediterranean or temperate zones.
- Management strategies. While forest management recommendations have been included, the authors could provide more concrete examples or elaborate on how their findings might influence post-fire conservation and restoration policies.

Overall, the responses are solid and well justified. A deeper exploration of these points could further enrich the manuscript and enhance its usefulness for researchers and land managers. Nevertheless, again, I am satisfied with the work made and the revised document.

(Remarks on code availability)

Reviewer #1

Thanks for the opportunity to review the manuscript “Exposure of western United States bird communities to predicted high severity fire.” The manuscript presents the results of a novel approach combining species distribution models and fire forecasting to assess the exposure of bird species to future fire regimes in western United States, identifying both areas and species of conservation and management concern.

The manuscript is well written and communicated, the analyses overall well justified and the results noteworthy. However, I have a number of concerns specifically relating to clarity of methodological details and important context in which the results need to be placed.

Thank you for these kind words, we address the overarching concerns and line edits below.

First, the methods make clear that the change in fire regime at the focus of this study is places likely to experience severe fire, should a fire occur in a location. This is a very important nuance to the paper that needs to be made clear throughout the introduction. Further, from reading the methods of the study referred to as the source of the fire severity predictions, it’s not clear how future fire severity is modelled given the Parks et al. 2018 study only provides predictions for the year 2016. The authors need to make clear whether they are using a different set of predictions from the Parks et al. model and describe the methods used to generate those (given the study has been described as giving predictions of future fire severity exposure, Line 58) or clarify the framing of the paper as being about ‘current’ fire severities if it’s the 2016 predictions being used. This is a very important part that needs to be clarified in future versions of the manuscript as it impacts how the results can be framed.

We agree that this is an important aspect of the fire severity data product that was not sufficiently discussed in the previous version of the manuscript. We used the data product as described in Parks et al. 2018, so yes, the predictions are based on 2016 conditions. Although we don’t use the word “current” in the manuscript, we see that this could be a source of confusion. We have now made explicit that the input data was from 2016 in the introduction: “Predictions are based on the climate and landscape conditions at the time the product was developed (2016)...”. We have also expanded our description of the data product and its limitations in the methods section:

“Predictions are based on variables that summarize live fuel, topography, climate, and fire weather for conditions in 2016. We do not expect that these variables have changed

significantly since being measured with the exception of live fuel, which is impacted by fires that have burned in the interim period. Rather than invalidating the hotspot maps, these discrepancy areas represent realizations of the refugia or area of concern designations.”

Future fire modelling is inherently uncertain, however the results presented here appear to have used the mean predictions of the fire severity modelling despite the likely reasonably large uncertainties that drive fire severity (e.g. variation in daily fire weather & other future conditions). Given the main results are locations on maps, I think that there needs to be an acknowledgement of the uncertainties in the two major analyses of the manuscript (i.e. the hotspot analysis and the species exposure analysis).

Predictions are based on the mean weather conditions under which fires occur, which is now explicitly stated in the methods section. We have also expanded our limitations section to acknowledge this potential source of uncertainty: “Fire predictions are also based on mean fire weather conditions despite a fair amount of variability in the conditions compatible with fire, adding some degree of uncertainty to our results, though overall findings are not likely to be substantially impacted.”

The majority of the uncertainty in the eBird relative abundance data product is around the relative abundance estimate rather than occurrence, which we convert to for all of our analyses. We do not therefore expect the locations of community-level biodiversity hotspots to be sensitive to this small amount of error for each species.

Given the lack of clarity in the manuscript about what conditions the predicted severities represent, it gets a little difficult to tell whether the authors are referring to historical or future severity values in different parts – need to make clear throughout (e.g. line 99).

We believe this point is now clarified after the text additions for the “first” point above.

Second, the manuscript focuses on the increase in bird species’ exposure of high severity fire but does not consider the relative vulnerability of each species in the analysis to changes in the amount of high severity fire beyond the example species identified in the discussion. This is acknowledged throughout, but given the hotspot analyses used, it’s highly plausible that the at risk hotspot locations would change if species’ vulnerability to severe fire was considered. I think that the authors need to at least acknowledge this.

We agree that vulnerability hotspots would likely be in very different locations. We were careful throughout the manuscript to use neutral language (e.g. “exposure”) to describe

the impacts of high severity fire as they vary substantially across species. With the exception of a few well studied species, we do not know whether most species would respond positively or negatively to high severity fire, and therefore cannot make comprehensive classifications of vulnerability. We've now added a statement in the first paragraph of the discussion making the distinction between biodiversity and vulnerability hotspots:

“As species responses to high severity fire are highly diverse, hotspots notably do not represent vulnerability, but rather areas where biodiverse communities face greatest exposure.”

Lastly, I think the functional traits analysis, while informative, needs a stronger justification for inclusion in the study. Why was this analysis done and why were these specific traits chosen? Is there a particular hypothesis the authors have about these traits in terms of vulnerability to severe fire? At present it's a little unclear if these traits were chosen based on data availability, rather than because they represented the major functional differences between species relevant to fire-related analyses.

Thank you for raising this concern which points to a gap in how we incorporated the trait analysis. In conjunction with responses to line comments below, we have now added language in both the results and methods sections to better outline our motivation for the analyses and trait selection. We view this as an exploratory analysis that may offer some insight into fire vulnerability that generates – rather than tests – key hypotheses for further research.

Line comments below

L50: Impacted? Need to replace with exposed as have not considered variability in vulnerability to high severity fire.

Now changed.

L58: Importance nuance here – this paper doesn't assess species exposure to future fire regimes, it assesses species exposure to high severity fire. Make sure this is consistent throughout.

We agree, thank you for catching this misleading wording, it is now changed: “to future fire” -> “to high severity fire”.

L63: Occurrence here, but relative abundance elsewhere (e.g. L237). I can see that different metrics are used at different points, but I think this needs to be made clearer through the manuscript. Especially when talking about the study's broader results.

We used the relative abundance data product from eBird Status and Trends, but since relative abundance is not directly comparable across species, community-based metrics were calculated based on occurrence only. We have now added a sentence to make this process and justification explicit in the "analysis" section of the methods, as well as changing "occurrence" -> "relative abundance" in L63 since we're referencing the data product here.

L68-69: Not clear in this sentence how the trait database was used – might need a second sentence here saying what was done with the trait database to pre-empt the methods.

This sentence has now been reworded to add detail about trait database use: "To calculate functional diversity, we paired all species in our study with species-level traits from the AVONET trait database describing morphology, habitat, and life history characteristics (Tobias et al., 2022)."

L71: Fire severity under what scenario of climate? At what time point in future?

Thank you for identifying the lack of clarity here about a critical aspect of the fire model, which assesses fire type likelihood based on the landscape conditions at the time of the data product's development. We have now added a sentence of additional description:

"Predictions are based on the climate and landscape conditions at the time the product was developed (2016) and assign high severity (i.e. stand-replacing in this system) or low severity fire as more likely if a fire were to burn today."

L80: To a non-American reader, may need to make it clear here or earlier where severity is introduced whether all high severity fire in all ecosystems in this study (western United States) is stand replacing or whether this statement needs to be nuanced a bit (i.e. are there some ecosystems where severe fire is not stand replacing?).

Good point, in this case the two terms are synonymous, which we clarified in the edit for the previous point.

L83: What is meant by top 30% of biodiversity distribution? Is this the top 30% of richness values? Or something else?

That's correct, we've now changed the language of the parenthetical for clarity: "(top 30% of the distribution for a given metric)".

L86: State the range of percentages in parentheses here

Now added.

L88: Why was 10% chosen as the threshold for a hotspot?

We have now added expanded justification to the text:

"The 10% most biodiverse watersheds for each ecoregion were considered hotspots, a threshold that allows for sufficient spatial aggregation for regional interpretation and accounts for range-restricted species more effectively than more restrictive cutoffs (Reid, 1998; Shrestha et al., 2019)."

L104: I'm not quite clear on what is meant by "high and low severity biodiversity distributions" here – do you mean high biodiversity values affected by high and low severity fire or something else?

Yes, it is now reworded for clarity: "high congruence between biodiversity distributions for high and low severity areas".

L107: Wording is a bit clunky here. Suggest including the word 'species' so it reads 'percentage of a species' total global population'

Now added for clarity, thank you.

L112: I would like to see a stronger justification of the traits analysis. Why is it important to look at which traits are more likely to be exposed to high severity fire?

We have now added a sentence elaborating on the motivation for using a trait-based approach:

"For insight into the characteristics of species disproportionately exposed to high severity fire, and potential areas of community-level vulnerability, we identified the traits related to the percentage of a species' population in the forested study area found in

expected high severity areas. We performed beta regressions with species traits as predictors and the population percent in high severity as the response variable for both breeding and nonbreeding populations.”

L123: How large an increase is this from the historical fire regime?

Unfortunately, we cannot assess this as the historical hotspots would also likely have varied in location, which we do not have the data to address.

L127-129: It would be good to give an example or two here from the results to help crystallise it for the reader.

Great point, we do have these details a couple paragraphs down after the methods and data are fully explained, and we feel that’s a better spot for the reader to appropriately interpret the geographic patterns. Let us know if you disagree. We have now referenced the figure in where you suggest in the opening paragraph to ground the assertion.

L152: I think it would be good to quantify this more formally. What is the proportional increase in high severity fire in these locations?

As the regions are more qualitative designations based on visual assessment of the map, we feel formally assessing the mismatch percentage at that level would be inappropriate. Percentages for the entire study area are currently depicted in Figure 3 and reported in the methods.

L185-186: This is key and I think could be drawn out a bit more in the manuscript. Is it possible that the hotspots could change locations if just focused on species particularly vulnerable to high severity fire, as distinct from the all species approach taken here.

We believe this comment is addressed by our response to the reviewer’s “second” main point above, as well as modifications to the text throughout that ensure neutral language (e.g. “exposure” to fire) about the impacts of fire on any individual species.

L190: This study referred to here is not in the reference list and wasn’t provided to reviewers.

This paper is now in print and the reference has been corrected.

L229-241: More detail needed here on which species were included. How many bird species have been modelled? What proportion of western United States bird species

overall? Over what time period were records collected from? I know some of this is included in the eBird references, but it's important context for this study and so should be included.

We have now added the number of species maps pulled and the number that occurred in the western United States and were therefore used in the study. We had maps for all species found in the study extent.

L243-246: Why were these traits chosen? Was this just based on what data were available? Were there other traits likely to be important that weren't captured by these traits that may change the functional traits space developed in lines L270-275?

While convenient that all traits were available in a single resource, we also feel confident that these traits are as complete a picture as possible, given available data, of the traits governing a species' ecological role. We have added an additional text describing the motivation behind trait selection:

“Together these traits describe the functional dimensions governing a species' ecological role as completely as possible for all birds included in our study. For birds, like most animal groups, general traits related to fire vulnerability or response have not been well identified, and we therefore cast a broad net to identify general associations (Geary et al., 2020; Pocknee et al., 2023).”

L246-255: I think that the fire severity predictions used in this study need further information. I understand that the authors reference the original study but there's important information in the original severity modelling studies that is important for understanding the present study here. For example, the abstract (e.g. L13) and introduction (e.g. L58) sets the paper up as being about assessing exposure to changing/future fire regimes, but it is not clear from the information presented here how the change from 'historical' fire regimes is being quantified. How does the Parks et al. studies predict future/changed fire severity? The methods in that study only describe making predictions to 2016 conditions. Has it since been used under some set of future conditions that are different from historical? Are these conditions in the near term, medium-term or long-term?

We have addressed some of these concerns in our response to the first general point above. The Parks data product tells us what the fire severity would be if it burned, and there is strong evidence in the existing literature (which we cite), that much of the high severity fire represents a novel fire regime. The data product itself does not tell us which predictions represent a novel fire type, which is why we incorporated the historical fire

regime data product. We are making the assertion that even these near-term forecasts of future fire largely represent already novel regimes, which we believe is well-supported by previous work. Hopefully this nuance is better communicated through the edits made in describing the data product and discussing its limitations.

L249: The 95% canopy mortality = stand replacing statement needs to be supported with a reference.

We have now added a citation (Miller et al. 2009).

L280: These thresholds need a stronger justification.

Please see the response above for the 10% hotspot threshold justification. The 20% forest statement was removed as it unintentionally reflected an old analysis approach not reflected in the figures. Thank you for prompting us to revisit it.

Figure 5: I think adding a dashed vertical line or similar to identify the mean/median of each density plot would be helpful, given the long tails of some of the distributions.

We agree, thank you for this suggestion, now added.

Reviewer #1 (Remarks on code availability):

Code was downloadable and included a README file. Appears to be enough information to re-run analysis in the Zenodo repository. I did not re-run the whole analysis though, I just viewed the code.

Reviewer #2:

Revision of the manuscript entitled “Exposure of western United States bird communities to predicted high severity fire” for Nature Communications

Thanks for the opportunity to review your manuscript. All of my comments and considerations below are intended solely to help improve the relevance of your manuscript, as I think your work is a great contribution to the fire ecology field.

First of all, I would like to congratulate you on the introduction, it is very well written, fluid, and well organized. Overall, your manuscript is of interest to wildfire specialists, ecologists, and others, even beyond your region. In this sense, however, this could also be a limitation, particularly in the introduction and discussion sections. From my point of view, it would be fantastic if these two sections were improved by incorporating a broader, more global perspective. Since understanding fire effects on biodiversity is a complex and widely studied challenge, I would encourage you to highlight how your work builds on or advances the recent research. To help on this, I suggest you consider referring to the following studies where appropriate:

1. Puig-Gironès, R., Palmero-Iniesta, M., Fernandes, P.M., Oliveras Menor, I., Ascoli, D., Kelly, L.T., Charles-Dominique, T., Regos, A., Harrison, S., Armenteras, D. and Brotons, L., et al., 2025. The use of fire to preserve biodiversity under novel fire regimes. *Philosophical Transactions B*, 380(1924), p.20230449. <http://www.doi.org/10.1098/rstb.2023.0449>
2. Kelly, L.T., Hoffmann, A.A., Nitschke, C.R., Pausas, J.G., Sanderfoot, O.V. and Tingley, M.W., 2025. Evolutionary Implications of Trait–Fire Mismatches for Animals. *Global Change Biology*, 31(7): p.e70368. <http://www.doi.org/10.1111/gcb.70368>
3. Puig-Girones, R., Brotons, L. and Pons, P., 2022. Aridity, fire severity and proximity of populations affect the temporal responses of open-habitat birds to wildfires. *Biological Conservation*, 272, p.109661. <https://doi.org/10.1016/j.biocon.2022.109661>
4. Arrogante-Funes, F., Aguado, I. and Chuvieco, E., 2024. Global impacts of fire regimes on wildland bird diversity. *Fire Ecology*, 20(1), p.25. <https://doi.org/10.1186/s42408-024-00259-x>

Thank you for your feedback on how we could make the manuscript more broadly applicable outside our region. Responses to some of the line comments below expand the geographic scope of the introduction, and we have now also incorporated some of your recommended citations to expand the scope of our bibliography. We do however feel that the framing of the introduction is as broad as possible while still allowing for the methods details required by the methods-last journal format. We do not, for example, reference our specific study region until the third paragraph of the introduction.

Secondly, I'm not comfortable evaluating your spatial modelisation, as it is outside my statistics expertise. Therefore, I have reviewed the section specifying the spatial model statistics, but I encourage the editors to find (if the other reviewers are also not familiar with this kind of spatial model) an expert in this specific spatial modelling approach to review that part. I know this may not be good news for the authors, but that is my honest opinion, and I prefer to be transparent about this important point.

Following are a few minor comments:

Line 67–68: Regarding the other two diversity metrics, could you provide more detail on what you mean by “a measure of the volume of the multi-dimensional trait space”?

Description is now reworded for clarity: “...functional diversity, which describes the trait space occupied by a given community.”

Line 83 – 85: Split the sentence.

Now changed to two sentences.

Line 88: Explain what “HUC12 watershed” means the first time that it appears in the manuscript.

We have now added additional description: “We identified biodiversity hotspots at the HUC12 watershed scale, the smallest hydrologic unit defining a basin’s drainage that is also a common unit for management assessments.”

Line 114: Use “Using these percentage” instead of “Using this percentage”.

Now changed.

Line 122 – 123: Although it is positive to focus on a specific region, this can reduce the perception of the global relevance of the work. At the same time, mentioning the applicability of the results to other parts of the world without providing examples limits the global applicability of the research. This is a pity. I know the focus is justifiable, but adding a comment about similarities with other regions could increase the impact. For example, there are other areas with similar types of ecosystems (Mediterranean regions and/or temperate forests) that could experience similar dynamics. In addition, a little

more information could be added about the implications of the impact of these changes. For example, “destroying habitats” is a very general, sensationalist, and somewhat exaggerated statement (given all the knowledge we have about wildfire ecology). It would be useful to specify which habitats and why they are so critical to biodiversity.

We agree that the relevance of our approach for other systems could be more explicitly illustrated. We have now added a number of examples and citations for systems experiencing similar regime shifts.

We agree that the phrase “destroying habitats” would be sensationalist, and we have refrained from using it anywhere in the text. Likewise, we have carefully used more precise terminology when discussing the process of high-severity, stand-replacing fire, which removes forest vegetation and can cause short-term abandonment by many forest-dependent species. We’ve now added the qualifier of time scale as shown in the text below, with the rest of the discussion outlining mechanisms by which it could remove habitat in the long-term (e.g. type conversion, maladaptation due to a mismatch with historical regimes).

“Up to 30% of forest bird diversity hotspots in the western United States are predicted to be exposed to future stand-replacing high severity fire, which could remove habitats for many forest-dependent species in the short and potentially long-term.”

Line 126 – 129: It would be useful to specify further how this exposure to low fire intensity may or may not benefit biodiversity. What types of refuges or protection mechanisms exist in these areas? How can this information help to design conservation strategies? I sincerely believe that you can get more out of this topic.

In this case we are using refugia in the ecological sense, rather than to reference a policy protection or potential for one. We have now added citations that discuss the term’s use in a fire context. We have also added a sentence in the introduction laying out why we think the “refugia” and “area of concern” designations are well-justified for their respective fire severities:

“These designations reflect strong evidence of low severity fire as a resilience mechanism and high severity fire as a destabilizing mechanism in these systems (Harris et al., 2021; Johnstone et al., 2016).”

We discuss the potential management and policy implications for refugia and areas of concern in the final paragraph of the discussion, including specific forest restoration approaches.

Line 133 – 137: Even with the reference to LandFire (2010), it would be useful to include more citations to recent or more detailed studies that reinforce the conclusions. It would also be good to compare it with data from regions outside the LandFire system to reinforce the conclusion. At the same time, the “mismatch” between historical and predicted regimes is important, but there isn't enough discussion about how this mismatch affects species or communities. Explaining how these discrepancies can have ecological, or conservation consequences would add some depth to the manuscript. Another consideration is that changes in fire regimes can quickly alter these perceptions. It would be good to highlight this through the bibliography. And if possible, management strategies to avoid adverse effects are also provided, this would be of great interest and help to managers.

We have now expanded our discussion of management strategies in the final paragraph of the discussion, including adding additional references. At this time we do not know of any other studies using this approach to look at biodiversity distributions and fire regime change, so we unfortunately cannot compare the prevalence of hotspots in new fire regimes to studies outside our study area. We do however feel the potential impacts of changing fire regimes are already well-discussed, with this paragraph and the second paragraph of the introduction dedicated to the topic.

Line 158 – 160: The discussion on habitat conversion lacks critical reflection on the methodological limitations of addressing this phenomenon. For example, the model may not be able to fully predict the long-term effects of habitat conversion, or there may be uncertainties in associating the effects of fires with changes in ecosystem type.

We do not directly assess the likelihood of post-fire type conversion in our analysis, but rather use this paragraph to speculate about which areas in our study may be most vulnerable to wholesale changes in their bird communities IF type conversion were to occur. This speculation is based on existing literature and studies of type conversion. Our text does not seek to weigh into debates on the merits or uncertainties surrounding type conversion – rather, we seek to compare our bird-focused results to existing information to highlight regions of the Western US that may warrant further attention.

Line 158 – 167: You need to better connect the implications for forest management and biodiversity conservation. How can the results be used to design specific management strategies for biodiversity conservation or post-fire restoration?

We now call out targeted management as an important measure to prevent type conversion, and elaborate on management approaches in the final paragraph of the discussion as it is a tool to address many of the issues raised in the discussion.

Line 164 – 166: The persistence of species (site tenacity) must be taken into account, which can last for one or two generations, a fact that could “limit” the loss of certain species in the short term.

We agree with this statement, but feel it is a level of detail unnecessary for a paragraph on long-term bird conservation in these landscapes.

It would also be important to consider the role of fire refuges (unburned patches) within burned areas that minimize the loss of certain species. It would be interesting to mention the need to maintain these refuges for wildlife as a post-fire strategy, especially in areas where high-intensity fires are expected. See for inspiration: <https://doi.org/10.1016/j.foreco.2023.121439>, <https://doi.org/10.1111/j.1469-1795.2012.00542.x> and their refs.

One of the limitations of our study is that we cannot talk about the impact of burn configuration on biodiversity outcomes, despite its significance in mediating fire impacts, as we do not have predictions of burn patterns. For example, we cannot say whether a high severity fire would be homogenous or patchy. We do not therefore include it in the discussion as our study cannot address landscape-level fire characteristics.

Line 173 – 176: It would be really helpful to give more context on why this conversion is important for biodiversity. What other studies or models support this prediction? Which species are most affected by this conversion? Keep in mind that there are species (mostly from open habitats) that do well with fires and also contribute to biodiversity.

This is an important nuance in the impacts of fire on birds (e.g., some birds have positive responses, some negative). Although our spatial, community-level analysis is forced to overlook this nuance via community metrics like richness and diversity, the follow-up single-species analyses (e.g., Figure 5) produce information that can be tied to known, species-level fire responses. The updated Discussion paragraph focuses on describing the patterns of biodiversity-severity overlap without assuming precisely how each species will respond.

In the case of the type conversion discussed in this paragraph, we agree that there should be more nuance in how different species in the community may be impacted, although we do think there’s evidence that type conversion can lead to overall species

richness loss. We have now added the following sentence with a new citation: “While some generalist species may be able to persist in the novel vegetation conditions, type conversion is a significant threat to forest-specialist species and can lead to loss of species richness in general (Vicini et al., 2025).”

At the same time, although there is a call for the need for large-scale assessments, some specific recommendations or guidelines could be provided for those responsible for forest management or biodiversity conservation. This would improve the usefulness of the study and its practical implications. See for inspiration <https://doi.org/10.1038/s43247-025-02165-9> and their refs.

We have now elaborated on our management recommendations in the final paragraph of the discussion, including adding additional references. We agree that implementation of management, particularly prescribed burns, is a complex ecological and social problem, but recommendations beyond the importance of thinning and prescribed burns to mitigate fire severity are too complex to comfortably fit in this manuscript.

Line 191 – 203: Good point, but some of my previous comments may help to improve these limitations and future section.

Thank you for your thoughts, by thoroughly addressing the previous comments we believe this limitations paragraph is now better grounded.

Line 208 – 210: Good point.

Thank you.

Line 224. How many 3 km x 3 km pixels you finally analysed after data curation?

Now added at the end of the first methods paragraph.

Line 228: Put a comma after “communities”: “To assess breeding and nonbreeding bird communities, we [...]”.

Comma now added.

Line 230 – 231: The inclusion of eBird’s data product from 2022, matched with bird trait data from AVONET, provides a strong foundation for modelling. However, missing species were accounted for using the 2021 version, which is reasonable given

computational limitations, but mentioning how any discrepancies between the two datasets were handled would strengthen this section.

Thank you, we have now added clarification that the occurrence and percent of population data we use in our analysis are robust to the minor method changes between versions:

“Since not all species were available for the 2022 version we used the 2021 version to fill species gaps, with species occurrence data and percent of population estimates both robust to slight methodology changes between data versions.”

Line 250: “non-transferable” instead “nontransferable”?

The sentence was removed over the course of responding to reviewers’ comments.

Line 242 – 255: The explanation of how data was adapted to account for ecoregion-specific fire probabilities adds robustness. This ensures that analyses across regions are comparable. However, more details on why these thresholds were selected, or how variable they are across ecoregions, would be helpful.

We provided additional text in this paragraph to provide more details on how and why the thresholds were selected [lines 276-293].

Line 274: Put a comma after “space”: “We used the first four PCA axes to describe the functional space, following best practices [...]”.

Comma now added.

Line 291 – 292: The use of upper- and lower-tail tests is a good approach to capture extremes in fire-severity impact. However, providing more details on the binomial tests’ parameters would make it easier for replicability transparency (e.g., how the expected proportions were set).

The parametrization is given by the ecoregion-level probability of a pixel being in a high severity area, which is describe a couple sentences prior. We have now added the parameters for the binomial test for each ecoregion and metric in the supplement to improve reproducibility.

Line 288: Put a comma after “ecoregion”: “For each ecoregion, we then performed [...]”

Comma now added.

Line 303 – 314: This approach is strong because it integrates ecological traits with exposure to high-severity fire, providing insights into which species are most vulnerable. The exclusion of non-forest species is reasonable, and the choice of traits (morphology, habitat, lifestyle) for regression modelling is appropriate. However, it would be helpful to provide more details on the beta regression models, such as the model assumptions and how AVONET traits were used as predictor variables.

We have now added a sentence on the assumptions and appropriateness of beta regression for modeling percentages, with a couple of citations. We also clarify that we used the untransformed AVONET traits as predictor variables.

Line 315 – 325: The cautious approach of treating the mismatch between historical and predicted fire regimes as conservative is good for transparency. However, the impact of using different severity thresholds on the comparison could be better articulated (i.e., how much of an overestimate is expected and why it matters). Moreover, more details on the specific methods used to compare these regimes (e.g., statistical tests or methods of comparison) would improve replicability.

Since the thresholds were not comparable between datasets, we did not perform any formal statistical tests, which we feel is a fair reflection of the data and prevents over interpretation.

Unfortunately, any statements about the degree of overestimation in historical fire severity would be extremely speculative, as the historical frequency of fires between 75% (landfire threshold) and 95% (our threshold) could range from very frequent to negligible. Our current statement about it simply being an overestimation is the strongest we feel is fair to the data.

Reviewer #1

The authors have done a great job responding to the previous round of reviews and suggestions. They have addressed all my queries adequately and the paper is much improved.

We are grateful for the opportunity to respond to the reviewers' thoughtful comments and agree that the paper has improved substantially.

I don't have anything more substantive to suggest or comment on with this latest version, beyond encouraging the authors be careful of the language in the manuscript regarding "changing fire regimes" given the additional detail provided about the fire severity model being from 2016 (so the paper is measuring bird species exposure to past changes in potential fire severities), rather than explicitly estimating future severities. 'Altered' or 'Changed' fire regimes may be more appropriate as those terms have a past tense and is a clearer reflection of the analyses actually done in this paper.

We agree that this is an important distinction for appropriate interpretation of our results. For the two places in the text where we use the phrase "changing fire regimes" to refer to our results, rather than findings of other studies or the concept in the abstract, we have now changed the phrase to "altered fire regimes".

Reviewer #1 (Remarks on code availability):

I reviewed the code in the first round of reviews.

Reviewer #2 (Remarks to the Author):

Revision of the manuscript entitled "Exposure of western United States bird communities to predicted high severity fire" for Nature Communications

Thanks for the re-opportunity to review your manuscript. Again, my comments and considerations are intended solely to help improve the relevance of your manuscript.

The authors have satisfactorily addressed most of the questions and comments raised by the reviewers, tackling methodological concerns, clarifying key points, and incorporating suggestions to enhance the clarity and relevance of the manuscript. In this regard, you have done a good job, and overall, I am satisfied with the revised document.

You have made significant adjustments to the text, including more detailed explanations of the data used, the limitations of the models, and the rationale behind the functional trait analyses. You have also expanded the discussion on the global implications of the study

and incorporated relevant references suggested by the reviewers.

We are grateful for the many thoughtful comments and are glad the previous round of revisions addressed all major concerns.

However, there are a few points that could benefit from further elaboration (apologies if I reiterate aspects from the initial review, we all have our biases and areas of expertise), and I encourage the authors to develop these more thoroughly in the text:

- Impact of habitat conversion. While the authors speculate on regions vulnerable to habitat conversion, they could delve deeper into the specific ecological implications of this phenomenon, including examples of how it affects forest and generalist species.

We believe that the process of type conversion is well described in the current text and that speculation, beyond the broad statement that generalist species may be able to persist where forest specialist cannot post-conversion, would be inappropriate given the large geographic and landcover scope of the analysis. To point to that nuance, we have now added a sentence in the type conversion discussion paragraph:

“The magnitude of species loss or turnover is likely dependent on how diverged the new habitat type is from pre-fire conditions and how specialized the pre-fire community was on the original habitat type.”

- Post-fire refugia. The authors mention that they are unable to address landscape-level burn patterns, but a more detailed discussion on the importance of unburnt refugia within fire-affected areas would be valuable. I encourage further development of this section, which I believe is fundamental and highly relevant to fire ecology and biodiversity.

We agree that fire refugia are an important mechanism for understanding biodiversity persistence in post-fire landscapes. We have now added a sentence to that effect in reference to the potential power of smaller scale active management:

“While treatment at the watershed-scale identified here may be too large to be feasible in some areas, even smaller restoration efforts can maintain unburned patches that act as refugia to support future biodiversity regeneration.”

We feel that further specific discussion of refugia in the context of the results may mislead readers as to the scale and implication of the data we used.

- Global applicability. Although examples of similar systems outside the study region have been added, the discussion could be expanded further to explore how the findings might be useful for other regions with comparable fire regimes, such as Mediterranean or temperate zones.

Since the two main findings of our study are 1) the identification of specific regions of concern, and 2) highly exposed species, both of which are heavily dependent on our region of interest, we do not feel comfortable speculating more on how patterns in the western United States may or may not translate to other regions. Particularly since the mechanisms conferring regional or species risk are not well known.

- Management strategies. While forest management recommendations have been included, the authors could provide more concrete examples or elaborate on how their findings might influence post-fire conservation and restoration policies.

We have elaborated some in reference to the refugia comment made above. We feel more detailed prescriptions for different regions are outside the scope of this paper and are based on both the ecological, managerial, and political context of specific areas.

Overall, the responses are solid and well justified. A deeper exploration of these points could further enrich the manuscript and enhance its usefulness for researchers and land managers. Nevertheless, again, I am satisfied with the work made and the revised document.

We thank you for your comments throughout the review process.